# Association between Chronic Pain and Physical Frailty in Community-Dwelling Older Adults

**DOI:** 10.3390/ijerph16081330

**Published:** 2019-04-13

**Authors:** Yuki Nakai, Hyuma Makizako, Ryoji Kiyama, Kazutoshi Tomioka, Yoshiaki Taniguchi, Takuro Kubozono, Toshihiro Takenaka, Mitsuru Ohishi

**Affiliations:** 1Department of Physical Therapy, School of Health Sciences, Faculty of Medicine, Kagoshima University, Kagoshima 890-8544, Japan; nakai@health.nop.kagoshima-u.ac.jp (Y.N.); kiyama@health.nop.kagoshima-u.ac.jp (R.K.); 2Graduate School of Health Sciences, Kagoshima University, Kagoshima 890-8544, Japan; reha_tommy@yahoo.co.jp (K.T.); p.taniguchi0601@gmail.com (Y.T.); 3Department of Cardiovascular Medicine and Hypertension, Graduate School of Medical and Dental Sciences, Kagoshima University, Kagoshima 890-0075, Japan; kubozono@cepp.ne.jp (T.K.); ohishi@m2.kufm.kagoshima-u.ac.jp (M.O.); 4Tarumizu Municipal Medical Center, Tarumizu Chuo Hospital, Kagoshima 891-2124, Japan; takenaka@tarumizumh.jp

**Keywords:** physical frailty, chronic pain, older individuals

## Abstract

This cross-sectional study investigated the association between chronic pain and physical frailty in community-dwelling older adults. We analyzed data obtained from 323 older adults (women: 74.6%) who participated in a community-based health check survey (the Tarumizu Study, 2017). Physical frailty was defined in terms of five parameters (exhaustion, slowness, weakness, low physical activity, and weight loss). We assessed the prevalence of chronic low back and knee pain using questionnaires. Participants whose pain had lasted ≥two months were considered to have chronic pain. Among all participants, 138 (42.7%) had chronic pain, and 171 (53.0%) were categorized as having physical frailty or pre-frailty. Logistic regression analysis showed that chronic pain was significantly associated with the group combining frailty and pre-frailty (odds ratio 1.68, 95% confidence interval 1.03–2.76, *p* = 0.040) after adjustment for age, sex, body mass index, score on the 15-item Geriatric Depression Scale, and medications. Comparing the proportions of chronic pain among participants who responded to the sub-items, exhaustion (yes: 65.9%, no: 39.4%) demonstrated a significant association (*p* < 0.001). Chronic pain could be associated with the group combining frailty and pre-frailty and is particularly associated with exhaustion in community-dwelling older adults. Therefore, there is a need for early intervention and consideration of the role of exhaustion when devising interventions for physical frailty in older individuals with chronic pain.

## 1. Introduction

Frailty is a decline in physiological ability with aging [1]. Cellular defects accumulate with age, creating a variety of disorders, including the loss of functional capacity [2]. Determinants of frailty can be considered according to domains (physical, psychological, and social) [3]. Frailty has multidimensional aspects, but its functional aspects are especially important in order to understand it [4]. Older individuals with frailty have an increased risk of negative health outcomes, such as falling, various disabilities, a lower quality of life, hospitalization, and mortality [1,5,6,7,8]. Therefore, early identification and assessment of community-dwelling older individuals with frailty is required to prevent progression to negative health states in an aging society.

In a growing number of older adults, there is a heavy comorbidity burden and a high rate of geriatric syndromes including chronic pain [9]. Chronic pain, which causes a rise in healthcare costs and a deterioration in the quality of life, is a common symptom among community-dwelling older adults [9,10,11]. Epidemiological studies reveal that the prevalence of chronic pain is high in community-dwelling older adults [10,11]. For example, Liberman et al. reported a high rate of chronic pain (55.2%) and geriatric syndromes (85.4%) as well as an association between them [9]. Previous studies have also reported an association between chronic pain and limitations in the activities of daily living [12,13] because of deterioration in physical functioning [12,14], poor psychological status [15,16], and low physical activity levels [17,18]. Therefore, it is imperative to identify older adults with chronic pain earlier and develop a means of preventing chronic pain in community-dwelling older adults.

It is known that chronic pain among community-dwelling older adults is a risk factor for worsening frailty [19]. However, the relationships between the early stages of physical frailty and chronic pain and the sub-items of physical frailty and chronic pain in older adults remain unclear. 

Examining the cross-sectional relationship between physical frailty and chronic pain may provide information that could be helpful in developing more effective strategies to prevent frailty from a multidimensional perspective. In this study, we examined the effect of chronic pain as a risk factor for the potential development of physical frailty, and investigated which components of frailty are more relevant to chronic pain among community-dwelling older adults.

## 2. Materials and Methods

### 2.1. Participants

The current cross-sectional study used data from the Tarumizu Study 2017. Details of this study have been reported previously [20]. Briefly, the Tarumizu Study 2017, which involved collaboration between Kagoshima University (Faculty of Medicine), the Tarumizu city office, and Tarumizu Chuo Hospital, was conducted in November and December 2017 as a community-based health check survey. The participants of this study were selected from about 3810 older people over the age of 65 living in Tarumizu, a city in Kagoshima Prefecture, Japan. They were recruited through local newspaper advertisements and community campaigns. This was a health survey that is essentially based on the city’s health examination; the criterion for inclusion was being able to walk on one’s own, while the criterion for exclusion was having already received certification for long-term care. Informed consent was obtained from all participants prior to their inclusion in the study, and the Ethics Committee of the Faculty of Medicine, Kagoshima University approved the study protocol (Ref. No. 170103).

### 2.2. Physical Frailty

The status of physical frailty was based on the consisting of five components: exhaustion, slowness, muscle weakness, low physical activity, and weight loss [1]. Exhaustion was defined when answering “yes” to the next question from Kihon Checklist which is a self health checklist developed by the Japanese Ministry of Health, Labor and Welfare [21]: “Have you ever felt tired for no reason in the last two weeks?” Slowness was defined using (<1.0 m/s) as the normal walking speed cutoff value [22]. Weakness was defined using the maximum grip strength by different sex cutoff values (<26 kg for men, <18 kg for women) [23]. Low physical activity level was defined by the following questions: (1) “Do you do light exercise or sports for your health?” and (2) “Do you regularly exercise or sport?” “No” participants in both questions were categorized into low activity levels [24]. Weight loss was evaluated if participants answered “yes” to the following question, “Have you lost more than 2 kg in the past six months?” [21]. Participants who did not have any of these characteristics were not physically frail; physically pre-frail had one or two characteristics; and physically frail had three or more characteristics [25,26].

### 2.3. Chronic Pain

In this study, low back pain and knee pain were defined as chronic pain. These are the leading causes of pain complaints due to functional limitations and disorders in the elderly in Japan and other countries [27,28]. We assessed chronic pain by a face-to-face interview using the following four questions: “Do you have low back pain at the present time?” (yes), “Has that pain endured more than two months?” (yes), “Do you have knee pain at the present time?” (yes), and “Has that pain endured more than two months?” (yes) [19,29,30]. Participants with either low back or knee pain, or both lasting more than two months, formed the chronic pain group; those with neither low back nor knee pain formed the non-pain group; and those with low back or knee pain for less than two months formed the acute pain group.

### 2.4. Covariates

Age (years), sex, body mass index (BMI), responses on the 15-item Geriatric Depression Scale (GDS-15) [24], and total medications used (number/day) were assessed and included as covariates. Regarding these conditions and medications, doctors and nurses interviewed directly.

### 2.5. Statistical Analysis

The Mantel-Haenszel tests were used to compare the trends in sex and chronic pain in the groups formed on the basis of physical frailty status (non-physically frail, pre-frail, and frail). A one-way analysis of variance for continuous measures was performed to assess differences among frailty status groups in age, BMI, GDS-15 scores, and the total number of prescribed medications. 

The association between physical frailty and the prevalence of chronic pain was examined using multivariate logistic regression analyses. Dependent variables were classified into two patterns: the group combining physical frailty and pre-frailty, or physical frailty alone. The model in the multivariate logistic regression analysis was adjusted for age, sex, BMI, GDS-15 score, and the number of prescribed medications as covariates. The adjusted odds ratios (ORs) for incidents related to physical frailty were estimated with 95% confidence intervals (95% CIs). The comparisons of the prevalence of physical frailty sub-items in participants and of their chronic pain were made using chi-square tests. All analyses were conducted using IBM SPSS Statistics 25.0 (IBM Japan, Tokyo, Japan). The level of statistical significance was set at *p* < 0.05.

## 3. Results

### 3.1. Subsection Characteristics of Participants

A total of 452 older individuals were enrolled in the Tarumizu Study 2017; 380 of them participated in a health check survey. The data from these 380 participants was eligible for the current investigation. In this study, participants aged under 65 years (*n* = 1, the survey was undertaken before the participant’s 65th birthday), with a history of diagnosis of dementia (*n* = 7), missing data on physical frailty assessments (*n* = 2), no response to questions on pain (*n* = 1), and grip strength measures for those with unsafe conditions (e.g., systolic blood pressure ≥ 180 mmHg) were excluded (*n* = 5). Participants with acute pain were also excluded (*n* = 41). Finally, data from 323 community-dwelling older adults (aged ≥ 65 years, mean age: 75.2 years, women: 74.6%) were analyzed (Figure 1).

The characteristics of the participants and the comparisons among groups divided by frailty status are described in Table 1. Out of 323 participants, 152 (47.1%) were considered physically non-frail, 152 (47.1%) were considered physically pre-frail, 19 (5.9%) were considered physically frail, and 138 (42.7%) met the criteria for chronic pain. Of the 152 non-frail participants, 52 (34.2%) had chronic pain; of the 152 pre-frail participants, 76 (50.0%) had chronic pain; and of the 19 frail participants, 10 (52.6%) had chronic pain. Regarding frailty status, there was a significant difference in chronic pain (*p* = 0.014). There were significant differences in certain characteristics of frailty status, such as age (*p* < 0.001), GDS-15 score (*p* < 0.001), and prescribed medications (*p* = 0.001), but no significant differences in BMI (*p* = 0.924) and sex (*p* = 0.226).

### 3.2. Associations of Prevalence of Chronic Pain and Physical Frailty

The results of logistic regression analysis using a group combining frailty and pre-frailty participants as dependent variables are shown in the following Table 2. In the crude models, chronic pain was significantly associated with a group combining frailty and pre-frailty (OR 1.95, 95% CI 1.24–3.05, *p* = 0.004). In the adjusted models, which included age, sex, BMI, GDS-15 score, and the number of prescribed medications as covariates, chronic pain (OR 1.68, 95% CI 1.03–2.76, *p* = 0.040) and GDS-15 score (OR 1.17, 95% CI 1.06–1.30, *p* = 0.003) was associated with a group combining frailty and pre-frailty. The results of logistic regression analysis using only the frailty group of participants as dependent variables are shown in the following. Chronic pain was not associated with only the frailty group in the crude models (OR 1.53, 95% CI 0.60–3.87) but also in the adjusted models including covariates (OR 2.83, 95% CI 0.79–10.21). Age (OR 1.22, 95% CI 1.09–1.36, *p* = 0.001), GDS-15 score (OR 1.50, 95% CI 1.20–1.87, *p* < 0.001), and the number of prescribed medications (OR 1.15, 95% CI 1.00–1.32, *p* = 0.047) were associated only with the frailty group in the adjusted models.

Analysis of the prevalence of chronic pain in participants corresponding to the sub-items that determine frailty is represented in Figure 2. Exhaustion (yes: 65.9%, no: 39.4%) was significantly associated with the prevalence of chronic pain (*p* < 0.001). Slowness (yes: 52.4%, no: 42.1%), weakness (yes: 50.0%, no: 40.2%), low activity (yes: 44.6%, no: 42.3%), and weight loss (yes: 51.1%, no: 41.4%) were not significantly associated with the prevalence of chronic pain.

## 4. Discussion

This cross-sectional study indicated that chronic pain in community-dwelling older adults could be associated with the group combining frailty and pre-frailty. That group was associated with chronic pain even after the adjustment for potential covariates. Also, participants with exhaustion had a significantly higher proportion of chronic pain.

The development of an effective prevention strategy for frailty in community-dwelling older adults is a pressing issue in a rapidly-aging society (e.g., Japan), as frailty has a negative effect on health [1]. Chronic pain among community-dwelling older adults is a risk factor for worsening frailty [19]. The prevalence of chronic pain in this study was almost the same as in a previous study investigating Japanese adults [29], which reported that 1032 (39.3%) adults (mean age 57.7 years) out of 2628 sampled had chronic pain. Makizako et al. investigated community-dwelling older adults using the same definition of physical frailty as used in the present study, and reported that 56.5% of older adults were frail or pre-frail [26]. Therefore, the present findings are consistent with previous results, and are therefore credible. The previously mentioned prospective cohort study reported that physical frailty, and even being pre-frail, strongly predicts an increased risk of disability in Japanese older adults [26]. In the present study, the presence of chronic pain in the physically frail or pre-frail groups was significantly higher than in the non-frail group. This may be explained by the fact that chronic pain has profound effects on a number of health outcomes [31]. Chronic pain is associated with and may be the origin of various health hazards, including depression and comorbidities that exacerbate physical frailty. Also, in this study, an association between frailty and depression is suggested. Thus, our findings suggest that interventions in chronic pain in the early stages of physical frailty (mostly pre-frail status) may have something to do with preventing the risk of developing disability in community-dwelling older adults.

Analysis of the prevalence of chronic pain and sub-items of frailty showed that exhaustion, not other sub-items, was significantly associated with chronic pain. Many previous studies have reported on the relationship between pain and physical function deterioration [12,14,32,33]. However, in the present study, chronic pain was associated with exhaustion rather than with physical function. Shega et al. [31] argued that older adults with persistent pain, lack of sleep, and poor nutrition may experience a decrease in their physiological reserves, which increases the likelihood of falls and cognitive dysfunction. Exhaustion may be related to sleep deprivation and nutritional status. This may cause mental distress and also lose confidence. Then, they may express feelings that they cannot fulfill their social roles sufficiently with families, friends and neighbors. This could lead to withdrawal indoors and lead to a decline in physical activity as well. This suggests that in older adults, pre-frailty or frailty is initially associated with exhaustion, and subsequently perhaps with a decline in physical function. Among older adults, it is plausible that the multidimensional nature of chronic pain has an impact on one’s physiological systems, reduces one’s physiological reserves, and decreases one’s ability to maintain homeostasis [34]. Thus, there is a need for early intervention and consideration of the role of exhaustion when devising interventions for physical frailty in older adults with chronic pain.

This study includes several limitations. First, while chronic pain was recognized as an important aspect of physical frailty, the roles of the cognitive and psychological domains were not considered. Also, other potential covariates, such as lifestyle, nutritional, and hormonal factors [35,36,37], that could be related to exhaustion, were not considered. Second, we studied the relationship between chronic pain and the prevalence of frailty using cross-sectional design, but statistical power was insufficient due to the small number of participants. We need more prospective study with more participants. Additionally, regarding the definition of pain, we asked questions about low back and knee pain, but did not consider the degree of pain. The prevalence of low back and knee pain is high in Japan and other countries [27,28], but other parts of the body and different degrees of pain may need to be considered. We also need to consider other chronic pain cut-offs. In this study, registered 452 older adults accounted for about 10% of the surveyed subjects in the city. Moreover, they were not randomly selected. In this field, conducting longitudinal studies would accumulate valuable evidence.

## 5. Conclusions

Community-dwelling older adults with chronic pain were associated with the group combining physical frailty and pre-frailty. In particular, exhaustion may be associated with the prevalence of chronic pain. The maintenance of a patient’s physically non-frail status through interventions accounting for exhaustion can be a strategy for the maintenance of physical function among community-dwelling older adults with chronic pain.

## Figures and Tables

**Figure 1 ijerph-16-01330-f001:**
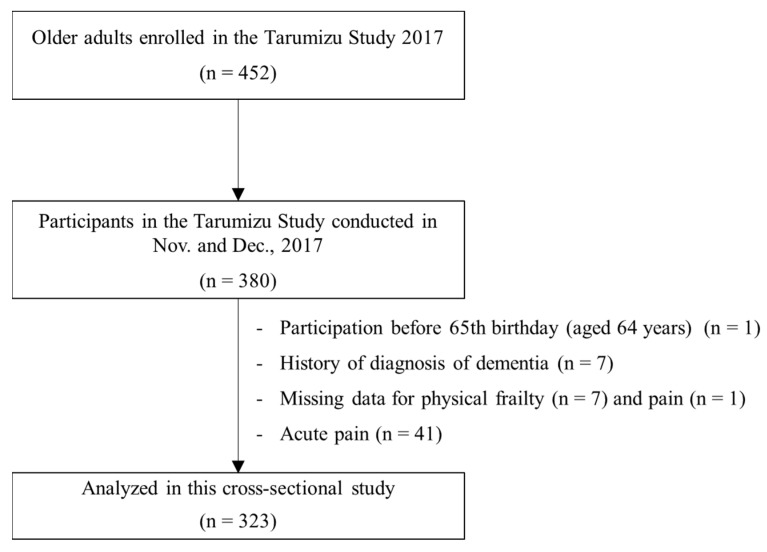
Participant inclusion criteria flow diagram.

**Figure 2 ijerph-16-01330-f002:**
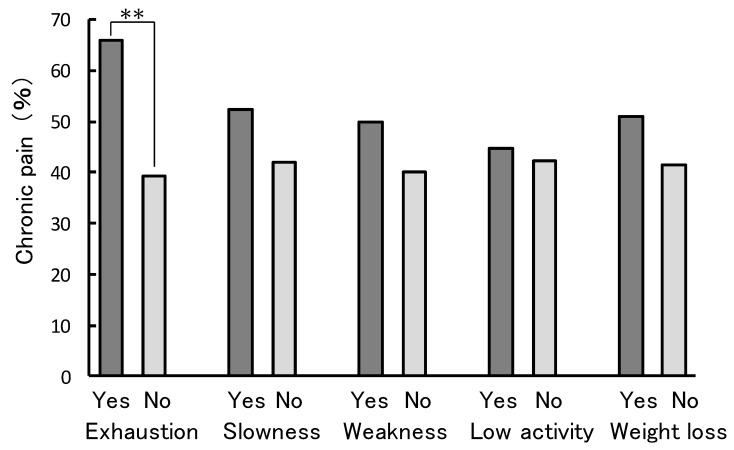
Association of the sub-items of physical frailty and presence of chronic pain. χ^2^ test for proportions; ** *p* < 0.01.

**Table 1 ijerph-16-01330-t001:** Characteristics of the participants (mean ± SD or %).

Variable	Total	Non-Frailty	Pre-Frailty	Frailty	*p* *
	(*n* = 323)	(*n* = 152)	(*n* = 152)	(*n* = 19)	
Chronic pain, n (%)	138 (42.7)	52 (34.2)	76 (50.0)	10 (52.6)	0.014
Age, mean ± SD (years)	75.2 ± 6.5	74.1 ± 5.8	75.4 ± 6.7	81.9 ± 7.1	<0.001
Women, n (%)	241 (74.6)	115 (75.7)	109 (71.7)	17 (89.5)	0.226
BMI, mean ± SD (kg/m^2^)	23.5 ± 3.4	23.4 ± 3.2	23.5 ± 3.5	23.6 ± 4.2	0.924
GDS-15 score mean ± SD (points)	2.5 ± 2.5	2.0 ± 2.0	2.8 ± 2.5	5.2 ± 3.5	<0.001
Medications mean ± SD (number)	3.7 ± 3.9	3.0 ± 3.4	4.1 ± 4.0	6.0 ± 5.3	0.001

SD, standard deviation; BMI, body mass index; GDS-15, 15-item version of the Geriatric Depression Scale. * Mantel-Haenszel test for proportion trends and one-way analysis of variance for continuous measures.

**Table 2 ijerph-16-01330-t002:** Logistic regression analysis between chronic pain and physical frailty status.

Independent Variable	Dependent Value: Classification of Two Patterns of Physical Frailty Prevalence Rates							
	Non-Frailty & Pre-Frailty vs Frailty				Non-Frailty vs Pre-Frailty & Frailty			
	Crude		Adjusted		Crude		Adjusted	
	OR (95% CI)	*p*	OR (95% CI)	*p*	OR (95% CI)	*p*	OR (95% CI)	*p*
Chronic pain	1.53 (0.60–3.87)	0.371	2.83 (0.79–10.21)	0.112	1.95 (1.24–3.05)	0.004 **	1.68 (1.03–2.76)	0.040 *
Age			1.22 (1.09–1.36)	0.001 **			1.03 (0.99–1.07)	0.176
Women			2.15 (0.42–10.90)	0.358			0.80 (0.47–1.38)	0.428
BMI			1.21 (1.00–1.46)	0.058			1.00 (0.94–1.08)	0.938
GDS-15			1.50 (1.20–1.87)	<0.001 **			1.17 (1.06–1.30)	0.003 **
Medications			1.15 (1.00–1.32)	0.047 *			1.06 (1.00–1.14)	0.069

Note: OR, odds ratio; CI, confidence interval; BMI, body mass index; GDS-15, 15-item version of the Geriatric Depression Scale; the bold typeface indicates statistical significance; ** *p* < 0.01; * *p* < 0.05; adjusted for age, sex, BMI, GDS-15 score, and number of medications.

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
