# Peer review of "Association between Chronic Pain and Physical Frailty in Community-Dwelling Older Adults"

_ijerph, 2019, doi:10.3390/ijerph16081330_

Round 1

Reviewer 1 Report

This article addresses an important topic; Chronic Pain and Frailty. In General it is well written. I've one major concern and some minor comments.  

Major concern

It is obvious that in this study a power problem occurred. The group of frail participants (n=19, 5.9%) is too small to result in a significant result in the logistic regression analysis. The solution to combine the groups of pre-frail and frail is just an artificial way of increasing the groupsize but does not allow to conclude that chronic pain is significant correlated to frailty status as stated in the conclusion. This problem has not been addressed properly throughout the manuscript. It should be at least an important issue in the discussion section and the conclusion should be more in line with this limitation.

Minor comments

Introduction

The introduction can be improved by adding more background information on the different viewpoints of Frailty and the development of these viewpoints. Starting from the physical biomedical viewpoint of Fried et al toward the accumulation of deficits model of Mitnitski et al and the more multidimensional models of for example Gobbens et al à a suggestion for reference is Junius-Walker et al 2018 The essence of Frailty: a systematic review and qualitative synthesis on frailty concepts and definitions. European Journal of Internal medicine 56; 3-10.

Please rephrase the opening sentence of the introduction. Line 36-37

“Frailty is a state that is vulnerable to the resolution of homeostasis after a stressor event with increasing age”

This sentence is difficult to read and in my opinion in the current form not correct.

Please rephrase the following sentence in the introduction

Line 56-57 “Examining the cross-sectional relationship between physical frailty and chronic pain may provide information that could be helpful in developing more effective strategies to prevent the multidimensional construct of frailty”

This sentence suggests that you want to prevent the multidimensional construct of frailty but I think your intention is to write “Examining the cross-sectional relationship between physical frailty and chronic pain may provide information that could be helpful in developing more effective strategies to prevent frailty from a multidimensional perspective”

Materials and Methods

This section needs some improvements. You briefly introduce the Tarumizu study 2017 and refers the reader to another paper in which this study has been described in more detail. However to really understand the results of the current study it is important to give the reader more information about the Tarumize study. For example the inclusion and exclusion criteria and the way eligible participants were approached. This is important as you state in line 203 in the discussion section that this sample was not randomly selected in which you suggest that there is possible selection bias.

Please also discuss the extra inclusion and exclusion criteria for the current study

Furthermore from line 68 onwards you describe results in the Material and Methods section. I would prefer that you start the result section with; “A total of 452 older individuals were enrolled in the Tarumizu Study 2017; 380 of them participated in a health check survey. The data from this 380 participants was eligible for the current investigation”  

Section 2.3

In the introduction you describe very well the current knowledge of chronic pain in elderly, However the choice for asking specifically about low back pain and knee pain fails. Can you elaborate on this.

Section 2.4

BMI is not mentioned as covariate in this section. Please add BMI to the covariates.  

Result section

I would prefer if you can elaborate a bit more on the adjusted models with p-values of all covariates in the model.

Line 140-142 you write; “However, chronic pain was not associated with  frailty status in the crude models (OR 1.53, 95% CI 0.60–3.87) or in the adjusted models including covariates (OR 1.80, 95% CI 0.58–5.58).”

The use of the word ‘However’ in this sentence is not accurate as it values your results. In the result section you should present the results plainly. Please change the wording in for example ‘Logistic regression analysis using only the group of frail participants as independent variable …….’ And discuss this result in the discussion section (see also my major concern)

Discussion

Line 157-159 You start the discussion section with the following: “This cross-sectional study indicated that chronic pain in community-dwelling older adults could be associated with physical frailty. In particular, physically frail or pre-frail statuses were independently associated with chronic pain even after adjustment for potential covariates.” à see my major concern. The second sentence “in particular……..covariates” is not true! This statement should be attenuated.

line 184 you state that ‘Exhaustion is multidimensional’. What do you mean with this statement

Conclusion

line 206 you state:

"Community-dwelling older adults with chronic pain were associated with physical pre-frailty or frailty status."

this is not true. It should be attunuated (see als major concern)

Author Response

Manuscript ID: ijerph-468890

Type of manuscript: Article
Title: Association between chronic pain and physical frailty in community-dwelling older adults

Major concern

Comment

It is obvious that in this study a power problem occurred. The group of frail participants (n=19, 5.9%) is too small to result in a significant result in the logistic regression analysis. The solution to combine the groups of pre-frail and frail is just an artificial way of increasing the groupsize but does not allow to conclude that chronic pain is significant correlated to frailty status as stated in the conclusion. This problem has not been addressed properly throughout the manuscript. It should be at least an important issue in the discussion section and the conclusion should be more in line with this limitation.

Response

We appreciate your pointing that out. In our manuscript, the classification was unclear and could not be analyzed and expressed correctly regarding frailty and pre-frailty. We have redefined the analysis as a grouped analysis of frailty and pre-frailty throughout the manuscript, and we have carefully addressed the over-interpretation of the conclusions. We added the lack of statistical power as one of the study’s big limitations to the Limitations section. In addition, as you suggested, we added BMI to the analysis and corrected the results. Each value has changed, but the statistical significance of the main part has not changed and it has not affected the argument.

Location in the text

p. 1, line. 25-28

Logistic regression analysis showed that chronic pain was significantly associated with the group combining frailty and pre-frailty (odds ratio 1.68, 95% confidence interval 1.03–2.76, p = 0.040) after adjustment for age, sex, body mass index, score on the 15-item Geriatric Depression Scale, and medications.

p. 1, line. 30-32

Chronic pain could be associated with the group combining frailty and pre-frailty and is particularly associated with exhaustion in community-dwelling older adults.

p. 3, line. 112-113

the group combining physical frailty and pre-frailty, or physical frailty alone.

p. 4, line. 147-153

The results of logistic regression analysis using a group combining frailty and pre-frailty participants as dependent variables are shown in the following. In the crude models, chronic pain was significantly associated with a group combining frailty and pre-frailty (OR 1.95, 95% CI 1.24–3.05, p = 0.004). In the adjusted models, which included age, sex, BMI, GDS-15 score, and the number of prescribed medications as covariates, chronic pain (OR 1.68, 95% CI 1.03–2.76, p = 0.040) and GDS-15 score (OR 1.17, 95% CI 1.06–1.30, p = 0.003) was associated with a group combining frailty and pre-frailty.

p. 6, line. 189-191

our findings suggest that interventions in chronic pain in the early stages of physical frailty (mostly pre-frail status) may have something to do with preventing the risk of developing disability in community-dwelling older adults.

p. 6, line. 201-203

Second, we studied the relationship between chronic pain and the prevalence of frailty using a cross-sectional design, but its statistical power was insufficient due to the small number of participants. We need more prospective studies with more participants.

Minor comments

Introduction

Comment

The introduction can be improved by adding more background information on the different viewpoints of Frailty and the development of these viewpoints. Starting from the physical biomedical viewpoint of Fried et al toward the accumulation of deficits model of Mitnitski et al and the more multidimensional models of for example Gobbens et al à a suggestion for reference is Junius-Walker et al 2018 The essence of Frailty: a systematic review and qualitative synthesis on frailty concepts and definitions. European Journal of Internal medicine 56; 3-10.

Response

We appreciate your insight and helpful suggestion. Accordingly, some sentences and references were added regarding the multidimensional aspects of frailty. We also rechecked the grammar and sentence structure, particularly in the Introduction section of the revised manuscript.

Location in the text

p. 1, line. 37-40 

Frailty is a decline in physiological ability with aging [1]. Cellular defects accumulate with age, creating a variety of disorders, including the loss of functional capacity [2]. Determinants of frailty can be considered according to domains (physical, psychological, and social) [3]. Frailty has multidimensional aspects, but its functional aspects are especially important to understand it [4].

Comment

Please rephrase the opening sentence of the introduction. Line 36-37

“Frailty is a state that is vulnerable to the resolution of homeostasis after a stressor event with increasing age”

This sentence is difficult to read and in my opinion in the current form not correct.

Response

We appreciate your comment. We deleted the sentence and replaced it with the following statement:

Location in the text

p. 1, line. 37-40 

Frailty is a decline in physiological ability with aging [1]. Cellular defects accumulate with age, creating a variety of disorders, including the loss of functional capacity [2]. Determinants of frailty can be considered according to domains (physical, psychological, and social) [3]. Frailty has multidimensional aspects, but its functional aspects are especially important to understand it [4].

Comment

Please rephrase the following sentence in the introduction

Line 56-57 “Examining the cross-sectional relationship between physical frailty and chronic pain may provide information that could be helpful in developing more effective strategies to prevent the multidimensional construct of frailty”

This sentence suggests that you want to prevent the multidimensional construct of frailty but I think your intention is to write “Examining the cross-sectional relationship between physical frailty and chronic pain may provide information that could be helpful in developing more effective strategies to prevent frailty from a multidimensional perspective”

Response

We rechecked the grammar and sentence structure, and replaced our sentence with the statement you suggested: “Examining the cross-sectional relationship between physical frailty and chronic pain may provide information that could be helpful in developing more effective strategies to prevent frailty from a multidimensional perspective.” Thanks to you, we were able to express sentences and meanings correctly.

Location in the text

p. 2, line. 59-61

Examining the cross-sectional relationship between physical frailty and chronic pain may provide information that could be helpful in developing more effective strategies to prevent frailty from a multidimensional perspective.

Materials and Methods

Comment

This section needs some improvements. You briefly introduce the Tarumizu study 2017 and refers the reader to another paper in which this study has been described in more detail. However to really understand the results of the current study it is important to give the reader more information about the Tarumizu study. For example the inclusion and exclusion criteria and the way eligible participants were approached. This is important as you state in line 203 in the discussion section that this sample was not randomly selected in which you suggest that there is possible selection bias.

Please also discuss the extra inclusion and exclusion criteria for the current study

Response

We appreciate your suggestion. We added more details about Tarumizu study 2017, as well as about inclusion and exclusion criteria.

Location in the text

p. 2, line. 70-74

The participants of this study were selected from about 3,810 older people over the age of 65 living in Tarumizu, a city in Kagoshima Prefecture, Japan. They were recruited through local newspaper advertisements and community campaigns. This was a health survey that is essentially based on the city's health examination; the criterion for inclusion was being able to walk on one’s own, while the criterion for exclusion was having already received certification for long-term care.

Comment

Furthermore from line 68 onwards you describe results in the Material and Methods section. I would prefer that you start the result section with; “A total of 452 older individuals were enrolled in the Tarumizu Study 2017; 380 of them participated in a health check survey. The data from this 380 participants was eligible for the current investigation”

Response

We appreciate your suggestion. We moved the following sentence and Fig 1. to the results section: “A total of 452 older individuals were enrolled in the Tarumizu Study 2017; 380 of them participated in a health check survey. The data from these 380 participants was eligible for the current investigation.” We were able to make it easier to understand.

Location in the text

p. 3, line. 122-124

A total of 452 older individuals were enrolled in the Tarumizu Study 2017; 380 of them participated in a health check survey. The data from these 380 participants was eligible for the current investigation.

Comment

Section 2.3

In the introduction you describe very well the current knowledge of chronic pain in elderly, However the choice for asking specifically about low back pain and knee pain fails. Can you elaborate on this.

Response

We appreciate your question. In studies that investigated factors associated with chronic musculoskeletal pain in community-dwelling older adults in Japan, low back pain (64.3%) shows the highest incidence, followed by knees (62.2%) and shoulders (53.1%). There were no significant differences among groups according to age, gender, height, or weight. “Prevalence and impact of pain among older adults in the United States: Findings from the 2011 National Health and Aging Trends Study” (2011) shows the same order. Thus, low back and knee pain are common sites of pain in the elderly in Japan and are the leading causes of functional limitations and disorders in the elderly. I have added the following two references and brief sentences. Although this study does not investigate shoulders and hips, etc., this is noted in the Limitations section on p. 6, lines 219–220, so please refer to our comments there.

Location in the text

p. 3, line. 109-111

In this study, low back pain and knee pain were defined as chronic pain. These are the leading causes of pain complaints due to functional limitations and disorders in the elderly in Japan and other countries [27,28].

p. 6, line. 218-220

Additionally, regarding the definition of pain, we asked questions about low back and knee pain, but did not consider the degree of pain. The prevalence of low back and knee pain is high in Japan and other countries [27,28], but other parts of the body and different degrees of pain may need to be considered.

Comment

Section 2.4

BMI is not mentioned as covariate in this section. Please add BMI to the covariates.

Response

We appreciate your detailed check. Adding BMI to covariates showed that although the values changed, the statistical significance of the main parts did not change and mostly did not affect the argument.

Location in the text

p. 3, line. 103

Age (years), sex, body mass index (BMI), responses on the 15-item Geriatric Depression Scale (GDS-15) [24], and total medications used (number/day) were assessed and included as covariates.

Result section

Comment

I would prefer if you can elaborate a bit more on the adjusted models with p-values of all covariates in the model.

Line 140-142 you write; “However, chronic pain was not associated with frailty status in the crude models (OR 1.53, 95% CI 0.60–3.87) or in the adjusted models including covariates (OR 1.80, 95% CI 0.58–5.58).”

The use of the word ‘However’ in this sentence is not accurate as it values your results. In the result section you should present the results plainly. Please change the wording in for example ‘Logistic regression analysis using only the group of frail participants as independent variable …….’ And discuss this result in the discussion section (see also my major concern)

Response

We appreciate your suggestion. We changed the sentences, the contents, and Table 2 as you suggested. We assume that this will make it easier to understand.

Location in the text

p. 4, line. 147-159, Paragraph 3.2.

The results of logistic regression analysis using a group combining frailty and pre-frailty participants as dependent variables are shown in the following. In the crude models, chronic pain was significantly associated with a group combining frailty and pre-frailty (OR 1.95, 95% CI 1.24–3.05, p = 0.004). In the adjusted models, which included age, sex, BMI, GDS-15 score, and the number of prescribed medications as covariates, chronic pain (OR 1.68, 95% CI 1.03–2.76, p = 0.040) and GDS-15 score (OR 1.17, 95% CI 1.06–1.30, p = 0.003) was associated with a group combining frailty and pre-frailty. The results of logistic regression analysis using only the frailty group of participants as dependent variables are shown in the following. Chronic pain was not associated with only the frailty group in the crude models (OR 1.53, 95% CI 0.60–3.87) but also in the adjusted models including covariates (OR 2.83, 95% CI 0.79–10.21). Age (OR 1.22, 95% CI 1.09–1.36, p = 0.001), GDS-15 score (OR 1.50, 95% CI 1.20–1.87, p < 0.001), and the number of prescribed medications (OR 1.15, 95% CI 1.00–1.32, p = 0.047) were associated only with the frailty group in the adjusted models.

p. 4, line. 156, Table2.

Table2. Logistic regression analysis between chronic pain and physical frailty status.

Independent Variable

Dependent   Value: Classification of Two Patterns of Physical Frailty Prevalence Rates

Non-Frailty &   Pre-Frailty

vs Frailty

Non-Frailty

vs Pre-Frailty & Frailty

Crude

Adjusted

Crude

Adjusted

OR (95% CI)

P

OR (95% CI)

P

OR (95% CI)

P

OR (95% CI)

P

Chronic pain

1.53 (0.60–3.87)

0.371

2.83 (0.79–10.21)

0.112

1.95 (1.243.05)

0.004**

1.68 (1.032.76)

0.040*

Age

1.22 (1.09–1.36)

0.001**

1.03 (0.99–1.07)

0.176

Women

2.15 (0.42–10.90)

0.358

0.80 (0.47–1.38)

0.428

BMI

1.21 (1.00–1.46)

0.058

1.00 (0.94–1.08)

0.938

GDS-15

1.50 (1.20–1.87)

< 0.001**

1.17 (1.061.30)

0.003**

Medications

1.15 (1.00–1.32)

0.047*

1.06 (1.00–1.14)

0.069

Note: OR, odds ratio; CI, confidence interval; BMI, body mass index; GDS-15, 15-item version of the Geriatric Depression Scale; the bold typeface indicates statistical significance; ** p < 0.01; * p < 0.05; adjusted for age, sex, BMI, GDS-15 score, and number of medications.

Location in the text

p. 6, line. 192-195

Also, in this study, an association between frailty and depression is suggested. Thus, our findings suggest that interventions in chronic pain in the early stages of physical frailty (mostly pre-frail status) may have something to do with preventing the risk of developing disability in community-dwelling older adults.

Discussion

Comment

Line 157-159 You start the discussion section with the following: “This cross-sectional study indicated that chronic pain in community-dwelling older adults could be associated with physical frailty. In particular, physically frail or pre-frail statuses were independently associated with chronic pain even after adjustment for potential covariates.” à see my major concern. The second sentence “in particular……..covariates” is not true! This statement should be attenuated.

Response

We appreciate your suggestion. When we analyzed it more carefully, as you suggested, we were able to sort out the association. We rephrased the sentence and attenuated the conclusion.

Location in the text

p. 6, line. 174-176

This cross-sectional study indicated that chronic pain in community-dwelling older adults could be associated with the group combining frailty and pre-frailty. That group was associated with chronic pain even after the adjustment for potential covariates.

Comment

line 184 you state that ‘Exhaustion is multidimensional’. What do you mean with this statement.

Response

We appreciate your question. We deleted it because it was a redundant expression.

Location in the text

p. 6, line. 202

Exhaustion is multidimensional and may be related to sleep deprivation and nutritional status.

Conclusion

Comment

line 206 you state:

"Community-dwelling older adults with chronic pain were associated with physical pre-frailty or frailty status."

this is not true. It should be attenuated (see als major concern)

Response

We appreciate your suggestion. We completely rephrased the manuscript and attenuated the conclusions. The conclusion now seems to be both easy to understand and accurate.

Location in the text

p. 7, line. 226

Community-dwelling older adults with chronic pain were associated with the group combining physical frailty and pre-frailty.

Reviewer 2 Report

Thank you for allowing me to review your manuscript. 

The manuscript is a secondary analysis of a community based cross sectional cohort study. The relationship between Frailty (Fried definition - modified) and chronic pain (knee/back pain for 2+ months) using univariate and multivariate analysis (meds, age, sex). Also investigated was the relationship between each of the 5 Phenotype Frailty criteria and chronic pain. Please see my suggestions below:

The manuscript in a few locations refers to the incidence/ development of frailty ('future development of frailty'). The final paragraph of the discussion acknowledges the inability to determine this in a cross sectional study; however, I do not think this is enough. It would be reasonable to say potential incidence/ development but it would be inaccurate to say anything stronger.

As for the 2 month cut off for back pain, the first time this is written in the manuscript, it includes reference 25. Two months very well may be long enough for pain to be considered chronic however ref 25 says " It is generally held that when pain persists beyond the expected timeframe for resolution and recovery from tissue injury, this constitutes the bridge from acute to chronic pain". A common back injury in general is a slipped disk (though I am not sure of the prevalence in >65 year olds). Pain with this typically can last up to 6 months before resolution.

Line 91, The labelling of the frailty group I presume was three or more and not more than three characteristics.

The paragraph beginning after '3.2' does not clearly explain table 2. I presume the crude model is referring to univariate analysis and the adjusted model is adjusted by sex, age, meds and GDS? Also, it would be clearer if after 'frailty status' on line 141 to add in 'alone'. It may also be helpful to have the first column (model) of table 1 to be described first in the text. 

Thank you

Author Response

Manuscript ID: ijerph-468890

Type of manuscript: Article
Title: Association between chronic pain and physical frailty in community-dwelling older adults

Comment

The manuscript in a few locations refers to the incidence/ development of frailty ('future development of frailty'). The final paragraph of the discussion acknowledges the inability to determine this in a cross sectional study; however, I do not think this is enough. It would be reasonable to say potential incidence/ development but it would be inaccurate to say anything stronger.

Response

We appreciate your suggestion. We rephrased the word and attenuated the contents. And, we rearranged the manuscript, as “incidence” and “prevalence” were mixed.

Location in the text

p. 2, line. 62

In this study, we examined the effect of chronic pain as a risk factor for the potential development of physical frailty, and investigated which components of frailty are more relevant to chronic pain among community-dwelling older adults.

Table2.

Dependent Value: Classification of Two Patterns of Physical Frailty Prevalence Rates

p. 6, line. 215-217

Second, we studied the relationship between chronic pain and the prevalence of frailty using cross-sectional design, but statistical power was insufficient due to the small number of participants.

Comment

As for the 2 month cut off for back pain, the first time this is written in the manuscript, it includes reference 25. Two months very well may be long enough for pain to be considered chronic however ref 25 says " It is generally held that when pain persists beyond the expected timeframe for resolution and recovery from tissue injury, this constitutes the bridge from acute to chronic pain". A common back injury in general is a slipped disk (though I am not sure of the prevalence in >65 year olds). Pain with this typically can last up to 6 months before resolution

Response

We appreciate your comment. This study used a two-month cut-off, so we will consider other cut-offs in the future. We added this to the Limitations discussion.

Location in the text

p. 6, line. 221

We also need to consider other chronic pain cut-offs.

Comment

Line 91, The labelling of the frailty group I presume was three or more and not more than three characteristics.

Response

We appreciate your detailed check. We rechecked the grammar and sentence structure, and rephrased it as you suggested.

Location in the text

p. 3, line. 89-91

Participants who did not have any of these characteristics were not physically frail; physically pre-frail had one or two characteristics; and physically frail had three or more characteristics [25,26].

Comment

The paragraph beginning after '3.2' does not clearly explain table 2. I presume the crude model is referring to univariate analysis and the adjusted model is adjusted by sex, age, meds and GDS? Also, it would be clearer if after 'frailty status' on line 141 to add in 'alone'. It may also be helpful to have the first column (model) of table 1 to be described first in the text.

Response

We appreciate your suggestion. We changed the sentences and the contents after '3.2' clearly. Table 2 was also changed, as other reviewers suggested. Each value has changed, but the statistical significance of the main part has not changed and it has not affected the argument.

Location in the text

p. 4, line. 143-154, Paragraph 3.2.

The results of logistic regression analysis using the group combining frailty and pre-frailty participants as dependent variables were shown in the following. In the crude models, chronic pain was significantly associated with the group combining frailty and pre-frailty (OR 1.95, 95% CI 1.24–3.05, p = 0.004). In the adjusted models, including age, sex, BMI, GDS-15 score, and number of prescribed medications as covariates, chronic pain (OR 1.68, 95% CI 1.03–2.76, p = 0.040) and GDS-15 score (OR 1.17, 95% CI 1.06–1.30, p = 0.003) were associated with the group combining frailty and pre-frailty. The results of logistic regression analysis using only the group of frailty participants as dependent variables were shown in the following. Chronic pain was not associated with only the group of frailty in the crude models (OR 1.53, 95% CI 0.60–3.87) and in the adjusted models including covariates (OR 2.83, 95% CI 0.79–10.21). Age (OR 1.22, 95% CI 1.09–1.36, p = 0.001), GDS-15 score (OR 1.50, 95% CI 1.20–1.87, p < 0.001), and number of prescribed medications (OR 1.15, 95% CI 1.00–1.32, p = 0.047) were associated with only the group of frailty in the adjusted models, respectively.

p. 4, line. 156, Table2.

Table2. Logistic regression analysis between chronic pain and physical frailty status.

Independent Variable

Dependent   Value: Classification of Two Patterns of Physical Frailty Prevalence Rates

Non-Frailty &   Pre-Frailty

vs Frailty

Non-Frailty

vs Pre-Frailty & Frailty

Crude

Adjusted

Crude

Adjusted

OR (95% CI)

P

OR (95% CI)

P

OR (95% CI)

P

OR (95% CI)

P

Chronic pain

1.53 (0.60–3.87)

0.371

2.83 (0.79–10.21)

0.112

1.95 (1.243.05)

0.004**

1.68 (1.032.76)

0.040*

Age

1.22 (1.09–1.36)

0.001**

1.03 (0.99–1.07)

0.176

Women

2.15 (0.42–10.90)

0.358

0.80 (0.47–1.38)

0.428

BMI

1.21 (1.00–1.46)

0.058

1.00 (0.94–1.08)

0.938

GDS-15

1.50 (1.20–1.87)

< 0.001**

1.17 (1.061.30)

0.003**

Medications

1.15 (1.00–1.32)

0.047*

1.06 (1.00–1.14)

0.069

Note: OR, odds ratio; CI, confidence interval; BMI, body mass index; GDS-15, 15-item version of the Geriatric Depression Scale; the bold typeface indicates statistical significance; ** p < 0.01; * p < 0.05; adjusted for age, sex, BMI, GDS-15 score, and number of medications.

Reviewer 3 Report

The prevalence of chronic pain is high in community- dwelling older adults and   it is known that chronic pain is a risk factor for worsening frailty. Examining the cross-sectional relationship between physical frailty and chronic pain may provide information that could be helpful in developing more effective strategies to prevent the  multidimensional construct of frailty. So the aim of the study was to examine the effect of chronic pain as a risk factor for the future development of physical frailty, and investigated which components of frailty  are more relevant to chronic pain among community-dwelling older adults.

The manuscript  is well written,  has a clear friendly structure (Introduction, Materials and Methods, Results, Discussion and Conclusions) and is easy to read. The subject is interesting and useful and  the paper raises important issues regarding the relationship between chronic pain and physical frailty in community-dwelling older adults.  The manuscript stands carefully developed methodology, good in- depth analysis of the results and comprehensive discussion containing the Limitation section. The text is complemented by  2 tables and 2 figures  and  enriched with 37 current references.

Minor issue:

1.      The study used data from the Tarumizu  Study 2017.  It would be worthwhile to know a little bit more about this study,  especially how looked the selection process of participants and how elderly people were selected into the study.

Author Response

Manuscript ID: ijerph-468890

Type of manuscript: Article
Title: Association between chronic pain and physical frailty in community-dwelling older adults

Comment

The study used data from the Tarumizu Study 2017. It would be worthwhile to know a little bit more about this study, especially how looked the selection process of participants and how elderly people were selected into the study.

Response

We appreciate your suggestion. We added more details about Tarumizu study 2017, and about inclusion criteria and exclusion criteria.

Location in the text

p. 2, line. 70-74

The participants of this study were selected from about 3,810 older people over the age of 65 living in Tarumizu, a city in Kagoshima Prefecture, Japan. They were recruited through local newspaper advertisements and community campaigns. This was a health survey that is essentially based on the city's health examination; the criterion for inclusion was being able to walk on one’s own, while the criterion for exclusion was having already received certification for long-term care.

Round 2

Reviewer 1 Report

Thank you very much for your thorough revision. I've no further comments

Reviewer 2 Report

Thank you for the edits to the manuscript, I believe it is a stronger manuscript as a result.

I have no additional comments.